# Life satisfaction and happiness in patients shielding from the COVID-19 global pandemic: A randomised controlled study of the 'mood as information' theory

Alice O'Donnell[1], Lydia Wilson[1], Jos A. Bosch[2], Richard Borrows[3]*

1 University of Birmingham Medical School, College of Medical and Dental Sciences, University of Birmingham, Birmingham, United Kingdom, 2 Faculteit der Maatschappij- en Gedragswetenschappen, Programmagroep: Clinical Psychology, Universiteit van Amsterdam, Amsterdam, The Netherlands, 3 Department of Nephrology and Kidney Transplantation, Queen Elizabeth Hospital Birmingham, Birmingham, United Kingdom

* Richard.Borrows@uhb.nhs.uk

**Data Availability Statement:** All relevant data are within the manuscript and Supporting Information files.

## Abstract

### Objectives

To extrapolate the 'mood as information' theory to the unique and ecologically relevant setting of the COVID-19 pandemic; the specific aim was to inform health care providers of the impact of bringing the pandemic to salience during life satisfaction evaluations, assessing whether this 'prime' results in increased or decreased reports of satisfaction which are derived unconsciously.

### Design

Prospective Randomised Interventional Study.

### Setting

Renal Transplant Department in a tertiary centre in the United Kingdom.

### Participants

200 Renal transplant patients aged between 20 and 88 years. Telephone interviews were undertaken between 1st May, 2020 and 29th May, 2020, at the height of 'shielding' from COVID-19.

### Interventions

Participants were randomised into 2 groups, with 1 group receiving a simple 'priming question' regarding the COVID pandemic and the other group having no prior contact.

### Main outcome measurements

Individuals were then asked to rate their own overall lifetime happiness; desire to change; overall life satisfaction and momentary happiness on a scale of 1 to 10 for each measure.

**Funding:** This work was funded by the renal department at the Queen Elizabeth Hospital. This research was independent from external funders.

**Competing interests:** The authors have declared that no competing interests exist.

Independent sample t-tests were used to compare results between the two groups, with a type 1 error rate below 5% considered statistically significant.

## Results

Participants' overall happiness with their life as a whole revealed that individuals who were primed with a question about COVID-19 reported *increased* overall happiness with their life compared to individuals who had not been primed (+0.88, 95% confidence interval 0.42 to 1.35, $p = 0.0002$). In addition, participants in the primed group reported *less* desire to change their life when compared to the non-primed group (-1.35, 95% confidence interval -2.06 to -0.65, $p = 0.0002$). Participants who were primed with the COVID-19 question also reported a *higher* overall satisfaction with their life than individuals who had not been primed (+1.01, 95% confidence interval 0.50 to 1.52, p = 0.0001). Finally, the participants who received the priming question demonstrated *increased* reported momentary happiness (+0.64, 95% confidence interval 0.03 to 1.24, $p = 0.04$).

## Conclusions

The results demonstrated that bringing salience to the COVID-19 pandemic with a simple question leads to positive changes in both momentary happiness and other components of global life satisfaction, thereby extrapolating evidence for the application of the mood-as-information theory to more extreme life circumstances. Given the importance of patient-reported evaluations, these findings have implications for how, when and where accurate and reproducible measurements of life satisfaction should be obtained.

## Introduction

The global coronavirus pandemic of 2019/2020 ('COVID-19') represents a thankfully rare, yet hugely salient event that had a negative impact on multiple aspects of people's lives across the world. This has particularly been the case for patients advised to undertake an extreme form of social isolation known as 'shielding', because of actual or perceived risk of disease severity. It became clear early in the pandemic, for example, that renal transplant patients should be included in the high-risk group [1], and were advised by the UK Department of Health to shield from COVID-19. Whilst appropriate on many levels, this unpleasant circumstance is likely to negatively impact upon life satisfaction and is inevitably accompanied by feelings of isolation, loneliness and frustration [2].

Life satisfaction and happiness are determined not only by overall personal dispositions and life-events, but also by momentary circumstances. In a landmark study—colloquially known as the 'weather' experiment—Schwarz and Clore identified the influence of such momentary circumstances on the judgement of individuals' lives more globally (in terms of satisfaction and happiness), and that this may occur outside a subject's awareness [3]. Specifically, these authors undertook telephone calls, asking participants for their subjective opinions on life satisfaction and happiness. The simple, but intriguing randomisation process involved asking participants: "how is the weather with you?" prior to the core life satisfaction questions. When the momentary mood-effect of poor weather (but not, in contrast, good weather) was made salient in this way, the negative effect of momentary mood on the judgement of one's life was removed (a so-called 'discounting effect') [4]. The theoretical implications of this finding

are twofold. Firstly, the study offered the 'mood as information' theory (also known as 'affect as information' theory) which predicts that transient alterations in mood also changes how individuals evaluate their life circumstances more globally. Secondly, by bringing plausible causes of negative mood (such as poor weather) into the awareness of the individual, the participants were provided with information they may use to attribute any negative feelings, allowing them to 'discount' the effect of their momentary mood on global judgement of life satisfaction [5, 6].

It is reasonable to suggest that such attributional and discounting effects could be extrapolated from weather to other scenarios. Specifically, the COVID-19 pandemic provides a unique example of a profound negative affective prime. We were, therefore, interested to evaluate its effects on life satisfaction and happiness in our renal transplant recipients undertaking 'mandatory' shielding, as an extreme model of the social isolation practices undertaken by many individuals across the world. This represented an opportunity to explore whether the above theoretical considerations and experimental findings may have real-life implications. If a patient's evaluation of satisfaction with their quality of life is to guide rational clinical decision-making, then understanding how and when such evaluations can be influenced by random circumstantial factors (that the patient is unaware of) becomes an important component of this strategy.

We utilised methodology derived from the Schwarz and Clore's (1983) study, described above, conducting telephone interviews with patients shielding at the height of the UK epidemic [3]. These patients were randomised to either receive or not receive, a simple priming question about the COVID-19 pandemic before answering questions about their life satisfaction. Based on extant theory, the use of this COVID-19 priming question could plausibly result in two mutually exclusive results. On one hand, raising awareness of the pandemic may bring attention to patients' negative situations, resulting in lower scores for overall life satisfaction [7]. Alternatively, the priming question may allow participants to attribute their mood to the negative external circumstance, and such explanatory discounting will yield higher scores for the subsequent life satisfaction questions.

Interestingly, and perhaps somewhat counter-intuitively, but nevertheless in agreement with the aforementioned 'weather study', our study demonstrated the latter hypothesis to be valid.

## Materials and methods participants

Two hundred-and-nineteen kidney transplant patients who were actively shielding from COVID-19 were invited to participate in the study, 19 of whom declined. Participants were recruited to this study from the renal transplant clinic at the Queen Elizabeth Hospital Birmingham, a large tertiary hospital serving the urban population of the UKs second largest city in which rates of COVID-19 were second only to London for the duration of the study. Specifically, and as part of a wider response to COVID-19 by medical students of the University of Birmingham, a student support worker was tasked with coordinating the uninterrupted supply of life-preserving immunosuppression medications to a number of prevalent renal transplant recipients. This was undertaken universally by telephone conversations. Once the clinical aspects of their care were discharged, patients were asked if they would be willing to participate in the study to evaluate their life satisfaction and wellbeing via a short telephone conversation the following week. The study was then undertaken between 1st and 29th May, 2020. To put this in perspective, these transplant patients (along with other vulnerable groups) received letters advising to shield on 21st March 2020; the UK was placed into 'lockdown' on 23rd March and easing of restrictions began on 6th July. Patients were advised to cautiously cease shielding from July 31st 2020.

The study was approved by the local clinical audit and research department of the Queen Elizabeth Hospital. In light of the circumstances, methodology and lack of identifiable patient-level information in the results, consent was implied by participation in the study, obtained verbally, and the requirement for signed consent was waived. The study was exempt from clinical trials registration on the basis it sought to inform providers' knowledge and attitudes, rather than influence or inform patients' health or behaviours. Patients were free to drop out of (or decline) the study at any time, and a minimum of 7 days elapsed between requesting participation and the subsequent telephone study itself, allowing time for full consideration regarding participation.

## Study design and data collection

Patient randomisation was undertaken using a random number generator which separated those to be primed with a question about the COVID-19 pandemic (n = 114), and those who were not (n = 86).

All participants were then asked the same 4 questions about their life satisfaction (see below) and their results recorded accordingly. The scripted conversation used the following introduction: "Hello, I'm ___ from the Queen Elizabeth Hospital. We spoke last week about a project we are doing within the renal transplant department about patients' life satisfaction and mood. Are you still willing to participate?"

For those agreeing to continue with the study (n = 200) at that point, patients in the 'primed' ('experimental') group (n = 114), were asked: "By the way, how's the COVID-19 pandemic making you feel at the moment?" The researcher did not engage in conversation in regard to this; rather, after the participant responded, the researcher replied with: "Well, let's get back to the research—we are interested in our patients' life satisfaction and mood as it is very important to us. Would you be able to answer 4 brief questions for us?" In the non-primed group (n = 86), the interviewer continued after the standard opening, without the question about the COVID-19 pandemic.

Following the appropriate introduction, participants were asked the following 4 questions, taken verbatim from the original 'weather' experiment described above and asked in the same order.

1. "First, on a scale of 1 to 10, with 10 being the happiest, how happy do you feel about your life as a whole?"

2. "Thinking of how your life is going now, how much would you like to change your life from what it is now? This is also on a scale of 1 to 10. Ten means "change a very great deal" and one "means not at all.""

3. "All things considered, how satisfied or dissatisfied are you with your life as a whole these days, with number 10 being the most satisfied."

4. "And, how happy do you feel at this moment? Again, 10 is the happiest."

The interviewer then concluded the telephone call with: "That's all the questions I have. Thank you for your time and cooperation".

Following the telephone call, patients' demographic and clinical data were retrieved from the departmental database and hospital electronic results system. The latest results prior to the lockdown period were recorded for analysis.

## Statistical analysis

All responses were integers between 1 and 10. Statistical Analysis was performed using Graph-Pad. Results were presented as mean ± standard deviation (SD). Independent sample t-tests

were used to compare results between the two groups, with a type 1 error rate below 5% considered statistically significant.

## Ethical approval

Protocol for this project was approved by the relevant CARMS committee (clinical audit and research) at the Queen Elizabeth Hospital.

## Results

The characteristics of the studied population are shown in Table 1. No differences in routine demographics or transplant-related metrics were evident between groups, with the characteristics typical of a prevalent renal transplant population. Specifically, Table 1 shows results for key parameters related to the transplant status and renal history which could conceivably be tied to life satisfaction and mood. Other routine biochemical and haematological data, evaluated for the last set of laboratory tests prior to the lockdown period, was also compared between groups, and no differences found.

The mean time from the start of the shielding period to the interview with participants was 51.5 ± 8.5 days, and the mean time from the interview till the end of the shielding period was 74.5 ± 8.5 days.

**Table 1. Summary of participant demographics.**

|  | COVID-19 primed group | Non-COVID-19 primed group |
|---|---|---|
|  | (n = 114) | (n = 86) |
| **Mean age (SD[1])** | 54.0 (12.3) | 54.2 (12.3) |
| **Gender (Male)** | 68/114 (59.6%) | 50/86 (58.1%) |
| **Race:** |  |  |
| White | 84/114 (73.7%) | 63/86 (73.2%) |
| South Asian | 24/114 (21.0%) | 20/86 (23.2%) |
| African-Caribbean | 6/114 (5.3%) | 3/86 (3.5%) |
| **Time on dialysis prior to transplant (IQR[2])** | 31 (2–70) months | 29 (2–71) months |
| **Diabetes:** |  |  |
| Pre-transplant | 13/114 (11.4%) | 10/86 (11.6%) |
| PTDM [3] | 17/114 (14.9%) | 14/86 (16.3%) |
| **Live Donor Transplant Recipient** | 20/114 (17.5%) | 17/86 (19.8%) |
| **Time Post Transplant (IQR)** | 52 (22–122) months | 51 (20–123) months |
| **Prior Acute Rejection[4]** | 17/114 (14.9%) | 15/86 (17.4%) |
| **Last eGFR[5] prior to shielding (SD)** | 45 (18) ml/min | 46 (17) ml/min |
| **Tacrolimus versus non-Tacrolimus based immunosuppression[6]** | 85/114 (74.6%) | 67/86 (77.9%) |
| **Body Mass Index (SD)** | 27.9 (5.6) kg/m$^2$ | 28.2 (5.7) kg/m$^2$ |

1: Standard Deviation (parametric data)

2: Interquartile range (non-parametric data)

3: Post Transplant Diabetes Mellitus

4: Acute transplant rejection was diagnosed by transplant biopsy in all cases; any episode at any time post-transplant (irrespective of histological or clinical classification) was captured

5: Estimated Glomerular Filtration Rate

6: Most 'non-Tacrolimus' treated patients received ciclosporin. A minority of patients were treated with either sirolimus or no calcineurin inhibitor at all (total n = 6 across both arms of study)

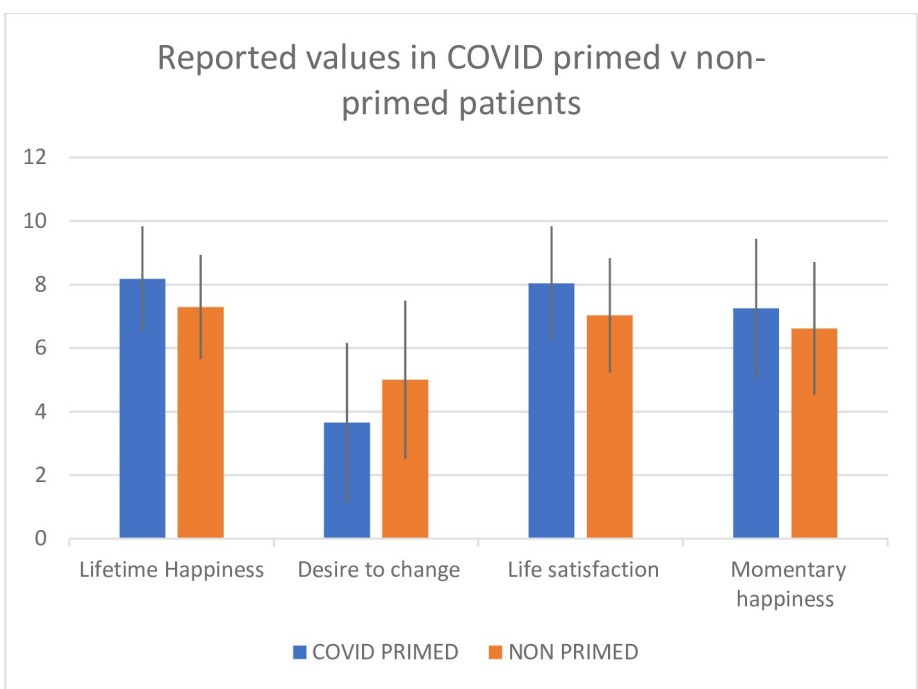

**Fig 1. Graphical representation of the results from Table 1, showing the mean reported scores with standard deviation bars.**

A summary of the results of the 4 questions in the primed and no-primed groups can be seen in Fig 1, with the full set of data shown in the S1 Data. The answers to all questions across the groups were normally distributed. Differences observed between the two groups in overall happiness ($t198 = 3.77$; $p = 0.0002$), desire to change ($t198 = 3.78$; $p = 0.0002$), satisfaction with life as a whole ($t198 = 3.93$; $p = 0.0001$) and happiness in this moment ($t198 = 2.08$; $p = 0.04$) were all significant.

## Discussion

The study is predicated on the basis that humans desire explanations for behaviours, thoughts and feelings. The mood-as-information theory predicts that knowledge of what influences one's momentary feelings (moods) is used to evaluate and re-evaluate other judgements about one's life. The present study sought to evaluate the influence of momentary mood upon global life satisfaction and overall happiness in the 'unique' setting of a viral pandemic among people (kidney transplant recipients) who underwent shielding. Asking patients a question about this negative event, was predicted to be utilised as an explanation for their mood, resulting in a discounting effect whereby 'momentary happiness' is used to explain, and adjust for judgements about overall happiness and satisfaction with life. In keeping with this, the results showed that bringing salience to the COVID-19 pandemic (with a simple question) results in positive changes in both momentary happiness and other components of global life satisfaction, thereby extrapolating evidence for the mood-as-information theory to these more extreme life circumstances with realistic ecological validity. Our observations refute the equally plausible prediction that bringing negative circumstances into awareness would prime or facilitate other negative evaluations, at least when the source of negative mood can be attributed 'externally' (see discussion below).

As mentioned in the introduction, this (mis)attribution phenomenon seems to be unidirectional; attribution of negative events will improve low mood, but the converse does not occur [8]. Indeed, it has been suggested this may represent a mechanism by which humans maintain a state of 'positive mood offset', which in turn promotes varied and coordinated cognitions in general [9]. Our findings do indeed resonate with this notion, although we propose that two other assumptions require attention before accepting the mechanism described above. Firstly, that the affective stimulus is a negative one. In this regard, Schwarz and Clore assumed that adverse weather represented a negative affective stimulus, and we consider that an assumption that the COVID-19 pandemic should be considered similarly is a reasonable one. Although at the time of writing, relatively little data has been released, emerging evidence does suggest a negative affective response to COVID-19 [10]. There has been unprecedented disruption of lives and work; health, distress and life satisfaction of working adults in China one month into the COVID-19 outbreak and further information will emerge as the pandemic progresses [11]. Secondly, to produce the results of the current study, negative affect requires attribution 'externally' and 'impersonally', that is to say they are attributed to the situation rather than the dispositional characteristics of the individual. In the face of negative events, people displaying a 'healthy' attributional style will also consider the events 'temporary' (i.e. not a permanent feature of their life moving forward), and not 'pervasive' (i.e. not representative of all facets of life) [12]. In summary, this style of attribution will result in unconscious processing along the lines of, "COVID-19 is a negative event for me, but not about me; won't be permanent and won't impact on every facet of my life". Assessment of attributional style was beyond the scope and logistics of the current study, and it is certainly possible that the cohort included patients with more pessimistic styles whereby events are characterised to be 'personal', 'permanent' and 'pervasive'. On the basis of the aggregate results, however, it would seem that the cohort as a whole displayed a broadly healthy/optimistic style of attribution. It is relevant in this regard to contrast the current results (and also those of Schwarz and Clore) with those of authors such as Strack et al (1988) who used a similar methodology but asked participants about their dating experiences before self-evaluation of life satisfaction [3, 6]. On this occasion, no effect of bringing relationship success into consciousness was seen, in all probability because relationship success will likely be regarded as 'personal', 'permanent', and 'pervasive', and therefore unamenable to external attribution. Participants in that study are likely to have viewed (from a psychological perspective) this information as a 'negative' prime in regard to overall life satisfaction, rather than as a source of explanation.

The results may have important clinical implications. In light of contemporary recognition of the importance of patient-reported evaluations, realisation that individuals' overall life satisfaction can be influenced by momentary circumstances, and that awareness of these influences may strongly determine such evaluations, has implications for how, when and where accurate and reproducible measurements of life satisfaction should be obtained. For example, self-reports of life satisfaction (either formal or informal) from patients before and after a (often arduous) four-hour haemodialysis procedure may differ due to how the individual attributes their feelings following the treatment, and whether the patient is made explicitly aware of the affective stimulus. Numerous situations exist whereby other stimuli, when brought to consciousness, may influence the subjective reports of life satisfaction; these may in turn influence patients' expectations, preferences, judgements and decisions. The field of 'clinical decision-making' has not taken these considerations into account historically, but rather focussed on how information is given to patients, and how questions and information are 'framed' [13]. We contend that consideration is required to consider the effect of bringing to salience experiences where affective information can be used to explain more durable life qualities and patient judgements. We believe the current study has demonstrated the former. The latter, we would argue, deserves more attention in the future. In essence, we propose that the affective

component of information be considered as important as the rational and reasoned components in the realm of joint decision-making. On a similar note, consideration of the attributional style of the patient, and how they might 'internalise' information is worthy of recognition. As discussed above, the COVID-19 pandemic seems to have been 'externally' attributed by the population overall, but inter-individual variation is likely and should be considered [14].

We should again clarify that by virtue of the study methodology, the participants will have no 'conscious' access to the reasons for their reports. In other words, merely asking the COVID question does not make them consciously more satisfied with life or happier, as they have no 'baseline' state to compare against. Rather the purpose of the study was to demonstrate the 'mood as information' effect to caregivers and health providers, in a relevant ecological context, such that they are aware of unrecognised influences on patient reports. A logical next step is to now use this theory to test whether patient-level decisions on their health or behaviour are modified by such influences. We strongly suspect this will indeed be the case, but this contention requires formal testing.

The raw scores that the patients in this study reported were of significant interest to us. We fully recognise that detailed assessment of multiple domains of life satisfaction and happiness were not obtained, but that was not the purpose of the study. Indeed, we consider this would, in fact, have weakened our ability to draw robust conclusions due to multiple measurements. Despite this study investigating patients with an established chronic disease, shielding from a global pandemic, it is striking that the average responses across the four domains of questioning are very similar to those seen from the Schwarz and Clore study undertaken 4 decades ago on another continent with a very different cohort [3]. Although early data does point to a reduction in life satisfaction during the pandemic [10, 11], further inspection shows that this may not necessarily be universal. Von Soest, for example, investigated only adolescents [10]. Of the adult population studied by Zhang, a proportion displayed no reduction in life satisfaction; this latter group being characterised by individuals exercising less than 2.5 hours per day [11]. It is certainly the case that many of the transplant recipients investigated in the current study would fall into that relatively sedentary group, thus our raw results do resonate with the findings of Zhang. Finally, it is widely accepted that healthy people over-estimate the impact of disease on sufferers [15]. We must not be ignorant of challenges faced by some patients in this study, and by extrapolation others with chronic health conditions, but we should similarly resist the paternalistic temptation to create a blanket burden of distress where none exists.

The results of this study provide a unique insight into the life satisfaction of patients shielding during the COVID-19 pandemic, and whose experiences may differ from other people. We describe the relationship between momentary mood and more global life satisfaction in the highest risk patients in society, helping to understand the nature and consequences of these relationships. We hope this will inform practice and expectations in regard to patient care during the current pandemic; during future emergencies and when the 'new normal' eventually emerges.

## Supporting information

**S1 Data.**
(XLSX)

## Acknowledgments

We would like to thank all patients for participating, and the members of staff in the renal department at the Queen Elizabeth Hospital for their efforts in supporting our work.

## Author Contributions

**Conceptualization:** Alice O'Donnell, Lydia Wilson, Richard Borrows.

**Data curation:** Alice O'Donnell, Lydia Wilson, Richard Borrows.

**Formal analysis:** Alice O'Donnell, Jos A. Bosch, Richard Borrows.

**Funding acquisition:** Alice O'Donnell, Richard Borrows.

**Investigation:** Alice O'Donnell, Lydia Wilson, Jos A. Bosch, Richard Borrows.

**Methodology:** Alice O'Donnell, Lydia Wilson, Jos A. Bosch, Richard Borrows.

**Project administration:** Alice O'Donnell, Lydia Wilson, Jos A. Bosch, Richard Borrows.

**Resources:** Alice O'Donnell, Lydia Wilson, Jos A. Bosch, Richard Borrows.

**Software:** Alice O'Donnell, Lydia Wilson, Jos A. Bosch, Richard Borrows.

**Supervision:** Richard Borrows.

**Validation:** Alice O'Donnell, Lydia Wilson, Jos A. Bosch, Richard Borrows.

**Visualization:** Alice O'Donnell, Lydia Wilson, Jos A. Bosch, Richard Borrows.

**Writing – original draft:** Alice O'Donnell, Lydia Wilson, Jos A. Bosch, Richard Borrows.

**Writing – review & editing:** Alice O'Donnell, Jos A. Bosch, Richard Borrows.

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
