## [Decision Letter · Decision Letter 0]

21 Sep 2020

PONE-D-20-25329

Life Satisfaction and Happiness in Patients Shielding from the COVID-19 Global Pandemic: A Randomised controlled study of the ‘Mood as Information’ Theory

PLOS ONE

Dear Dr. O'Donnell,

Thank you for submitting your manuscript to PLOS ONE. After careful consideration, we feel that it has merit but does not fully meet PLOS ONE’s publication criteria as it currently stands. Therefore, we invite you to submit a revised version of the manuscript that addresses the points raised during the review process.

The recommendations for changes are attached below.  As you will see these are minor.  Please submit your revised manuscript by Nov 05 2020 11:59PM. If you will need more time than this to complete your revisions, please reply to this message or contact the journal office at plosone@plos.org. Please include the following items when submitting your revised manuscript:

We look forward to receiving your revised manuscript.

Kind regards,

Itamar Ashkenazi

Academic Editor

PLOS ONE

Journal Requirements:

"This work was funded by the renal department at the Queen Elizabeth

Hospital. This research was independent from external funders."

"No. The funders had no role in study design, data collection and analysis, decision to publish, or preparation of the manuscript."

Reviewers' comments:

Reviewer's Responses to Questions

**Comments to the Author**

1. Is the manuscript technically sound, and do the data support the conclusions?

Reviewer #1: Yes

2. Has the statistical analysis been performed appropriately and rigorously? 

Reviewer #1: Yes

3. Have the authors made all data underlying the findings in their manuscript fully available?

Reviewer #1: Yes

4. Is the manuscript presented in an intelligible fashion and written in standard English?

Reviewer #1: Yes

5. Review Comments to the Author

Reviewer #1: Title: Life satisfaction and happiness in patients shielding from the COVID-19 Global Pandemic: A randomized controlled study of the ‘Mood as Information’ Theory

Study type: prospective randomized survey

Authors’ methodology and main results: The authors interviewed by telephone 200 kidney transplant patients shielding from COVID-19. They evaluated the effect of priming, as suggested by Schwarz and Clore’s study, on happiness and life satisfaction. Differences in happiness and satisfaction were noticed between those who were primed and those who were not.

Reviewer’s comments: I enjoyed reading this article. It asked a simple question and provided a simple answer. The discussion provides possible implications of priming on life satisfaction and happiness of our patients. I therefore recommend accepting this study for publication.

There are several minor issues that need to be addressed.

1. Results are repeated three times: within the text, in a table form and in a figure. This is unnecessary. I personally would prefer showing figure 1. The description within the text can then be shortened as follows:

A summary of the results of the 4 questions in the primed and no-primed groups can be seen in Figure 1. The answers to all questions across the groups were normally distributed. Differences observed between the two groups in overall happiness (t198=3.77; p=0.0002), desire to change (t198=3.78; p=0.0002), satisfaction with life as a whole (t198=3.93; p=0.0001) and happiness in this moment (t198=2.08; p=0.04) were all significant.

2. In figure 1, the authors provide the standard error. The authors should have drawn the standard deviation instead. Using the stars system with explanation provided just below the graph, the readers can understand which differences between columns were significant: * <0.05; ** <0.01; *** <0.001.

6. PLOS authors have the option to publish the peer review history of their article (what does this mean?). If published, this will include your full peer review and any attached files.

Reviewer #1: No

---

## [Author Response · Author response to Decision Letter 0]

3 Nov 2020

Response to reviewers:

Dear Reviewers, 

Thank you for your correspondence and for considering our study for publication. We are delighted that you agree our study holds merit and have provided us with such helpful comments. As a group we have met and discussed your comments in detail and worked through them in depth, hopefully addressing each point. We have revised the manuscript accordingly and would like to resubmit the revised version for your consideration. Below we have written our response to each of the comments provided. 

Thank you again for your helpful comments and we look forward to hearing from you soon.

Journal requirements

Set 1: 

Thank you for clarifying the style requirements PLOS ONE requires. In response to this comment we have altered the formatting to ensure that it complies with your requirements. We hope this satisfies your request.

"This work was funded by the renal department at the Queen Elizabeth

Hospital. This research was independent from external funders." We note that you have provided funding information that is not currently declared in your Funding Statement. However, funding information should not appear in the Acknowledgments section or other areas of your manuscript. We will only publish funding information present in the Funding Statement section of the online submission form.

Please remove any funding-related text from the manuscript and let us know how you would like to update your Funding Statement. Currently, your Funding Statement reads as follows: "No. The funders had no role in study design, data collection and analysis, decision to publish, or preparation of the manuscript."

Thank you for this point. In reply to this, we have removed the funding statements from within the text. Please update our funding statement to the following: ‘This work was funded by the renal department at the Queen Elizabeth Hospital. This research was independent from external funders’. In addition, our cover letter now says: ‘This work was funded by the renal department at the Queen Elizabeth Hospital. This research was independent from external funders’. 

We hope this is satisfactory and meets your request.

Thank you very much for this comment. Due to ethical restrictions we are unable to share this de-identified data set, with our data containing sensitive patient information. This is inline with the CARMS (Clinical Audit & Registration management System) ethics committee at the Queen Elizabeth Hospital. The CARMS committee are in charge of organising the ethics that a project must follow, and set out that for this project we must not share de-identified patient data set, given the sensitive nature of this study. To contact the CARMS department at the Queen Elizabeth Hospital, you can email clinicalaudit@uhb.nhs.uk . We have included details of this in the cover letter, as requested. We are happy that all relevant data is within the manuscript we have resubmitted. 

Thank you for pointing this out and we fully appreciate this comment. In response to this, we believe it simpler to simply state the lack of difference in these parameters (as one would expect in a randomised study) rather than subsume the reader in unnecessary data, either in the article itself or supplementary. This was agreed between all authors to be the best approach, and we hope this meets with your approval. We have removed the phrase “data not shown” that relates to this data in response to your comment.

We thank you for clarifying this for us. We have now only included an ethical statement in the methods section of our paper. 

Set 2

1. Results are repeated three times: within the text, in a table form and in a figure. This is unnecessary. I personally would prefer showing figure 1. The description within the text can then be shortened as follows:

A summary of the results of the 4 questions in the primed and no-primed groups can be seen in Figure 1. The answers to all questions across the groups were normally distributed. Differences observed between the two groups in overall happiness (t198=3.77; p=0.0002), desire to change (t198=3.78; p=0.0002), satisfaction with life as a whole (t198=3.93; p=0.0001) and happiness in this moment (t198=2.08; p=0.04) were all significant.

Thank you very much for this comment. We have replaced the summary of the results to comply with the above recommendation. Thank you again for this suggestion.

2. In figure 1, the authors provide the standard error. The authors should have drawn the standard deviation instead. Using the stars system with explanation provided just below the graph, the readers can understand which differences between columns were significant: * <0.05; ** <0.01; *** <0.001.

Thank you very much for this comment. Figure 1 has been changed to reflect the standard deviation rather than the standard error. This figure has replaced the previous figure and we hope you are happy with this change.

Thank you for notifying us about this tool, we have now uploaded the files to PACE.

---

## [Editor Report · Decision Letter 1]

19 Nov 2020

Life Satisfaction and Happiness in Patients Shielding from the COVID-19 Global Pandemic: A Randomised controlled study of the ‘Mood as Information’ Theory

PONE-D-20-25329R1

Dear Dr. O'Donnell,

We’re pleased to inform you that your manuscript has been judged scientifically suitable for publication and will be formally accepted for publication once it meets all outstanding technical requirements.

Kind regards,

Itamar Ashkenazi

Academic Editor

PLOS ONE

---

## [Editor Report · Acceptance letter]

24 Nov 2020

PONE-D-20-25329R1 

Life Satisfaction and Happiness in Patients Shielding from the COVID-19 Global Pandemic: A Randomised controlled study of the ‘Mood as Information’ Theory 

Dear Dr. O'Donnell:

I'm pleased to inform you that your manuscript has been deemed suitable for publication in PLOS ONE. Congratulations! Your manuscript is now with our production department. 

Kind regards, 

on behalf of

Dr. Itamar Ashkenazi 

Academic Editor

PLOS ONE